# Autonomic Nervous System in Obesity and Insulin-Resistance—The Complex Interplay between Leptin and Central Nervous System

**DOI:** 10.3390/ijms22105187

**Published:** 2021-05-14

**Authors:** Benedetta Russo, Marika Menduni, Patrizia Borboni, Fabiana Picconi, Simona Frontoni

**Affiliations:** 1Unit of Endocrinology, Diabetes and Metabolism, S. Giovanni Calibita, Fatebenefratelli Hospital, 00186 Rome, Italy; benedetta_russo6@msn.com (B.R.); marika.menduni@gmail.com (M.M.); borboni@uniroma2.it (P.B.); fabipicco@gmail.com (F.P.); 2Department of Systems Medicine, University of Rome Tor Vergata, 00133 Rome, Italy

**Keywords:** autonomic nervous system (ANS), leptin, insulin-resistance, central nervous system (CNS)

## Abstract

The role of the autonomic nervous system in obesity and insulin-resistant conditions has been largely explored. However, the exact mechanisms involved in this relation have not been completely elucidated yet, since most of these mechanisms display a bi-directional effect. Insulin-resistance, for instance, can be caused by sympathetic activation, but, in turn, the associated hyperinsulinemia can activate the sympathetic branch of the autonomic nervous system. The picture is made even more complex by the implicated neural, hormonal and nutritional mechanisms. Among them, leptin plays a pivotal role, being involved not only in appetite regulation and glucose homeostasis but also in energy expenditure. The purpose of this review is to offer a comprehensive view of the complex interplay between leptin and the central nervous system, providing further insights on the impact of autonomic nervous system balance on adipose tissue and insulin-resistance. Furthermore, the link between the circadian clock and leptin and its effect on metabolism and energy balance will be evaluated.

## 1. Introduction

Excessive weight gain is associated with the presence of cardiovascular risk factors, such as dyslipidemia, impaired glucose tolerance (IGT) and type 2 diabetes mellitus (T2D), hypertension, and kidney failure [1]. The increasing prevalence of obesity is a worldwide emergency, being associated with increased morbidity and mortality [2].

Neural mechanisms have been involved in the pathogenesis of obesity and insulin resistance, particularly sympathovagal imbalance, and the relative prevalence of sympathetic activity has been suggested to play a pivotal role in this complex bi-directional relationship [3]. Among the numerous mechanisms linking obesity and insulin-resistance (IR) with the imbalance of the autonomic nervous system (ANS), leptin has been widely investigated for its major role on energy expenditure regulation and sympathetic activation, mediated by its actions in the brain, possibly by increasing sympathetic outflow from the dorsomedial hypothalamus (DMH) [4].

The purpose of this review is to offer a comprehensive view of the complex interplay between leptin and the central nervous system (CNS), providing further insights into the impact of ANS balance on adipose tissue and IR. Furthermore, the link between the circadian clock and leptin and its effect on metabolism and energy balance will be evaluated.

## 2. The Bi-Directional Relationship between Autonomic Nervous System and Obesity/Insulin-Resistance

The ANS plays a central role in either the short-term or in the long-term regulation of body weight. Particularly, the short-term regulation of body weight is mainly mediated by the sensation of satiety, through gastric distension, due to the activation of vagal afferent nerves and gut hormones release. However, vagal activity is also involved in this latter pathway, since the activation of vagal afferents has now been recognized, as an early step in the control exerted by gut hormones on nutrient delivery to the intestine, by regulating food intake and gastric emptying (Figure 1) [5]. Therefore, gut hormones and vagal afferent neurons have been considered to play an important role in the pathogenesis of obesity [6].

More complex is the bi-directional relationship between obesity and sympathetic activity. In insulin-resistant states, such as in obesity, increased basal sympathetic activity has been reported and correlated with the degree of IR [7,8,9], suggesting that the reduced thermogenesis and the low metabolic rate observed in obese patients will eventually result in IR and compensatory hyperinsulinemia.

The consequent activation of the sympathetic nervous system (SNS) is associated with important hemodynamic effects on blood vessels, heart and kidney.

The stimulatory action of insulin on SNS activity is directly exerted in the brain: in the fasting state, low levels of plasma insulin reduce insulin-mediated glucose metabolism in hypothalamic neurons, resulting in the activation of an inhibitory pathway that suppresses chronically active sympathetic centers in the brain stem. Following carbohydrate intake, the increased insulin concentration stimulates insulin-mediated glucose metabolism in the same neurons, leading to augmented insulin-mediated glucose metabolism, reduction of the inhibitory pathway, and finally stimulation of the sympathetic centers at the brain-stem levels, with a consequent increase in central sympathetic outflow [3]. This mechanism has been proposed to explain the ‘‘pro-hypertensive’’ effect of insulin in susceptible individuals [10,11], where hypertension could represent the unwanted consequence of a compensatory mechanism recruited in the obese to restore energy balance and limit further weight gain (i.e., IR) (Figure 2). In other words, obese subjects, while resistant to the effects of insulin on peripheral glucose uptake, should not be resistant to the effect of insulin on the SNS, although this is not invariably associated with increased blood pressure due to a counterbalance of vascular compensatory mechanisms [12]. Obese subjects of the normative aging study were shown to be sensitive to the effects of insulin on sympathetic activity despite resistance to the effects of insulin on glucose uptake and displayed an increased 24-h urinary norepinephrine excretion [13], the amounts of norepinephrine excreted being related to the degree of obesity [14]. More recently, however, in adult obese patients, a blunted post-prandial increase in sympathetic tone has been demonstrated [15]. This interesting finding, though not univocally accepted [16], could represent a mechanism of inhibition of post-prandial thermogenesis, thus favoring weight gain.

The direct correlation between muscle basal nerve sympathetic activity and body fat suggests a link between SNS activity and obesity-related IR. Therefore, the prolonged stimulation of the sympathetic system, exerted by hyperinsulinemia in obesity, evokes hemodynamic responses, such as increased heart rate and cardiac output and reduced heart rate variability [17], which may represent the substrate for both the increased ischemic heart disease and the increased incidence of arrhythmias and sudden deaths, observed in obese patients.

However, a chronic increase in sympathetic outflow has been reported to decrease β-adrenergic responsiveness itself [8,18] through a down-regulation of the β-adrenergic receptors, which are known to mediate energy expenditure both at rest and after food intake. This mechanism could result in a reduced ability to dissipate energy and, as a consequence, a tendency toward weight gain.

In conclusion, in the complex regulation of body weight, a pivotal role is played by the ANS, both by the parasympathetic and the sympathetic branch. While the afferent vagal pathways seem to represent the most important link between the gut and the brain, sympathetic activation is involved in lipolysis increase and energy expenditure in white and brown adipose tissue (WAT and BAT), where, however, it results ineffective, in obesity. Sympathetic activation may favor the development of hypertension and organ damage in obesity and lead to overt cardiovascular disease.

## 3. The Interplay between Leptin and Central Nervous System Impact on Autonomic Effect on Obesity and Insulin-Resistance

Leptin is a hormone released by the WAT and has been shown to increase energy expenditure, acting both on the cardiovascular system and BAT thermogenesis via the arcuate nucleus (ARC) of the hypothalamus [19]. The ARC seems to represent the main site of action of leptin for stimulating SNS activity, as demonstrated by the lack of activation of sympathetic activity by leptin following ARC destruction [20]. However, animal studies have shown that leptin administration increases sympathetic outflow to the kidneys, the adipose tissue, the skeletal muscle vasculature and adrenal glands, also in different areas of CNS [21], thus resulting in augmented energy expenditure [22], and in sympathetic vasomotor activity [23].

Numerous data support the concept that leptin represents an important regulator of regional sympathetic nerve activity with pathophysiological implications in obesity [24].

Leptin regulates energy balance by decreasing appetite and increasing energy expenditure through sympathetic stimulation [25]. Leptin plasma levels decrease during fasting and increase after overfeeding [26]. It has been shown that rodents and humans that lack leptin or its receptor present dramatic hyperphagia with weight gain, whereas leptin administration demonstrated to decrease food intake [27,28].

It has been widely reported that leptin increases the sympathetic outflow to the kidneys, skeletal muscle vasculature and adrenal gland. A study conducted in non-obese rats reported that intravenous or intracerebroventricular (ICV) infusions of leptin increase sympathetic activity in the kidneys and adrenal glands [29]. According to these data, in non-obese Sprague-Dawley rats, intracarotid artery infusion of leptin for 12 days significantly increased mean arterial pressure and heart rate [30]. In addition, to explore the hypothalamic pathway of sympathetic activation of leptin, a study conducted on rats investigated the effect of leptin on circulating catecholamines and found that leptin administration caused a significant and dose-dependent increase in plasma concentration of norepinephrine and epinephrine [31]. These data suggest a relevant effect of leptin on arterial blood pressure.

Obesity is associated with circulating hyperleptinemia as a consequence of leptin resistance, suggesting that obese subjects have resistance to the anorectic and weight-reducing effects of leptin. At the same time, elevated plasma leptin levels may increase blood pressure and contribute to the development of hypertension [32].

Several studies confirmed the link between leptin and hypertension, reporting increased leptin levels in obese hypertensive individuals in comparison to obese normotensive subjects. Moreover, leptin deficiency in humans was associated with obesity and metabolic syndrome, without SNS activation or hypertension. These data confirm a role for leptin-mediated sympathetic activation in the pathogenesis of hypertension in obesity [5]. Conversely, sympathetic overactivity appears to be ineffective in favoring energy expenditure and, therefore, weight loss. Studies conducted on agouti obese mice with hyperleptinemia demonstrated that the anorexic and weight-reducing effects of leptin were attenuated compared to lean mice, while the effects of leptin on renal SNA were preserved [33]. This phenomenon of attenuation of the metabolic action (satiety and weight-reducing action) and preservation of the sympathetic actions of leptin was also observed during brain intracerebroventricular administration of leptin [34]. Therefore, the ability of leptin to influence cardiovascular sympathetic nerve hyperactivity seems to be unaltered in obesity [35] while being ineffective in regulating energy homeostasis.

Based on these findings, Mark et al. suggested a “selective leptin resistance”, in at least some forms of obesity, limited to its metabolic effects (satiety and weight loss), while it retains its sympatho-excitatory effects on the cardiovascular system via the SNS, through actions in the brain region of the DMH (Figure 3). This phenomenon might partially explain how hyperleptinemia can be associated with obesity but still contribute to sympathetic overactivity and hypertension because of the preservation of its sympathetic actions [36].

More recently, the evaluation of the differential effects of acute and chronic leptin stimulation on thermogenesis and lipolysis [37] but also on glucose metabolism [38] has gained much interest.

The activation of SNS by leptin occurs after food intake or cold exposure and induces lipolysis in WAT and heat production in BAT. In particular, sympathetic activation results in mobilization from WAT of free fatty acids (FFA), which are then used by BAT to dissipate energy as heat (Figure 4) [5]. Studies conducted on wild-type (WT) and leptin-deficient ob animals demonstrated that leptin treatment leads to a rapid depletion of fat mass [39]. In addition, the leptin-deficient ob/ob mice display low body temperature and do not survive acute cold exposure [40]. These data suggest a lipolytic and thermogenic effect of leptin respectively in WAT and BAT through the activation of SNS, which is not maintained in obese subjects [41]. Moreover, leptin has also been suggested to increase the formation of beige fat (Beige AT) through the browning of WAT, which contributes to adaptive thermogenesis and body fat reduction (Figure 4). Therefore, a failure in leptin action may contribute to the decrease of beige adipose tissue formation with a consequent reduction of energy expenditure [42].

Following the above, leptin levels reflect nutritional availability [43] and it exerts metabolic and thermogenic effects not only by changing sympathetic neural activity but also by dynamically regulating the architecture of the sympathetic nervous structure in adipose tissue. These effects of leptin on innervation in fat are mediated by the action of agouti-related peptide (AgRP) and pro-opiomelanocortin (POMC) neurons in the hypothalamic arcuate nucleus (ARC), as demonstrated by the reduced innervation in fat, following the deletion of the gene encoding the leptin receptor in either population. These AgRP and POMC neurons act via brain-derived neurotropic factor-expressing neurons in the paraventricular nucleus (PVN) of the hypothalamus, depending on its ability to modulate the level of SNS innervation of WAT and BAT. Moreover, several studies demonstrated that POMC neurons activated by leptin contribute to the normal regulation of glucose metabolism [38].

Indeed, leptin was shown to contribute not only to normal body weight regulation but also to the physiological control of blood glucose levels. It has been recently reported that leptin may improve glucose homeostasis and insulin sensitivity by increasing peripheral tissue glucose uptake and reducing liver glucose production [38]. Therefore the leptin resistance in obese subjects may contribute to developing IR. Moreover, several studies highlight a regulation of leptin secretion by insulin; it has been widely demonstrated in vivo that insulin stimulates leptin secretion from adipose tissue by transcriptional or post-transcriptional mechanisms [44]. At the same time, leptin has been demonstrated to directly affects pancreatic beta cell gene expression and lead to decreased insulin secretion. These data suggest the presence of crosstalk between leptin and insulin with regulatory feedback, which results altered in obese and IR subjects [45] (Figure 5) with a consequent relevant effect on glucose metabolism and sympathetic activity. Interestingly, a study that evaluated a sympathovagal balance in insulin-sensitive and IR subjects during the day life reported a sympathetic prevalence during the night in IR subjects compared to insulin-sensitive group highlighting the relevant role of the compensatory hyperinsulinemia on the sympathetic activity in the night time [46]. Therefore, these findings can suggest a subsequent hyperleptinemia associated to hyperinsulinemia which may contribute to the enhanced sympathetic activity and risk of hypertension development.

The link between energy balance and fertility has also been investigated. It has been demonstrated that leptin may influence the neural circuits related to reproduction. The study of Hill et al. reported that female mice lacking both leptin and insulin receptors in POMC neurons exhibit lengthened reproductive cycles, follicular arrest, hyperandrogenemia, and are sub-fertile [47]. These data suggest that POMC neurons may be the direct target of leptin and insulin actions important for fertility.

We may conclude that leptin signalling regulates the plasticity of the sympathetic architecture of adipose tissue via a top-down neural pathway that is crucial for energy homeostasis. However, the picture of the mechanisms involved in the pathogenesis of obesity is more complex, and within the neural circuity regulating energy balance, the participation of non-neuronal cells, such as glia, has been recently demonstrated [48]. The interactions of a high-fat diet, leptin signaling, and neuroinflammation have been described in the medial hypothalamus to be relevant to energy balance regulation. Indeed, leptin-deficient Ob/Ob mice exhibit profound hypothalamic gliosis when maintained on a high-fat diet [49].

## 4. Effect of Circadian Clock on Metabolism and Energy Balance: The Link between Leptin and Circadian Control

The circadian clock plays a crucial role in many biological processes and is organized in the central clock located in the suprachiasmatic nucleus (SCN) of the hypothalamus and in the peripheral clocks located in peripheral tissues (adipose tissue, liver, skeletal muscle and digestive tract). The light is the main synchronizer for the SCN, which in turn transmits timing signals to the peripheral clocks. In addition, other stimuli such as hormones, nutrients, feeding/fasting state, sleep-wake state, physical activity can affect the circadian rhythm in peripheral tissues [50].

The circadian clock is the main regulator of metabolism and energy homeostasis, and its disruption may lead to metabolic disorders and contribute to overweight and obesity. In animal models, SCN alterations, as well as clock gene mutations, result in dyslipidemia, insulin-resistance and hyperglycemia. In humans, circadian disorders such as deterioration of the sleep–wake cycle due to insufficient sleep, shift work and social jet lag have been shown to be associated with symptoms of the metabolic syndrome, including an impaired glucose tolerance and insulin sensitivity, hypertriglyceridemia and an increase in body mass index (BMI) and mean arterial blood pressure [51].

The presence of circadian clocks in WAT and BAT has been reported, and several pieces of evidence showed that the secretion of leptin is characterized by a circadian rhythm with serum leptin peak levels occurring during the night in mice and humans [52]. It has been extensively highlighted that the circadian clock plays a critical role in controlling the leptin endocrine feedback loop to maintain the homeostasis of energy balance [53]. In the CNS, the SCN clock rhythmically transmits the signal to the peripheral adipose tissue clock and potentiates the response of ARC neurons to circulating leptin (Figure 6). Studies conducted in rodents demonstrated that the ablation of the SCN eliminates leptin circadian rhythmicity suggesting that the central circadian clock regulates leptin expression. In white adipocytes, the BMAL1/CLOCK heterodimer that received signal from SCN directly controls leptin expression by regulating the activity of CCAATenhancer-binding protein alpha (C/EBPα), the most potent transcriptional activator for leptin (Figure 6) [54,55]. Therefore, a disruption of the circadian rhythm can affect leptin secretion with a consequence on metabolic process and energy balance. According to these data, research that used the mouse jet-lag model to stimulate circadian dysfunction demonstrated that chronic jet-lag resulted in high and arrhythmic serum levels of leptin. These results have been correlated with a weight gain and increase in fat mass independently of changes on the diurnal profiles of food intake or physical activity [54]. Importantly, the same research reported that leptin may affect the central circadian control directly via its receptor in ARC via POMC neuron and that circadian dysfunction, including chronic jet lag, leads by itself to leptin resistance. In addition, a recent study conducted in mice exposed to a constantly shifting lighting environment in order to chronically disrupt their circadian timing system demonstrated a decrease in central leptin signaling in light cycle-disrupted mice as indicated by a reduction in the number of phosphorylated signal transducer and activator of transcription 3 (STAT-3) immunoreactive cells in the ARC of the hypothalamus. Furthermore, cycle-disrupted mice displayed a significant increase in fasting blood glucose and showed an increase in body weight [56]. These data suggest that chronic light cycle disruption leads to altered leptin and insulin signaling, which may explain the association between circadian dysfunction and metabolic disorders and weight gain. In addition, it has been shown that high-fat diet (HFD) feeding also disrupts behavioral and molecular circadian rhythms, including eating behavior, locomotor activity, and expression of circadian clock genes [57]. In fact, a recent study conducted on mice examined the effect of HFD feeding on leptin signal transduction throughout the day; the results showed a decrease in leptin sensitivity in HFD mice compared to low-fat diet (LFD) mice. Furthermore, the authors provide evidence that HFD-induced leptin resistance is a temporary phenomenon and occurs only at specific intervals during the 24-h cycle suggesting that restricting food intake to leptin-sensitive time periods may be beneficial for metabolic health [51].

Interestingly, studies conducted on human adults showed an increase in food intake when sleep is acutely restricted, and food is provided ad libitum [58]. It has been reported that acute sleep restriction can lead to a decrease in leptin secretion and an increase in ghrelin plasma levels, promoting weight gain in healthy adults [59]. Moreover, it has been demonstrated that shifting from an insufficient to adequate sleep program decreased night food intake and led to weight loss [58]. These findings suggest that increased food intake during insufficient sleep is a physiological adaptation to provide the energy needed to support additional wakefulness; however, when food is easily accessible, food intake exceeds the energy needed. Recently, a study conducted on healthy participants investigated the effect of insufficient sleep on fasting metabolic hormones, and the results showed no difference in fasted leptin between participants with chronic sleep restriction and controls. These findings suggest that sleep restriction may have a limited impact on fasted concentrations of leptin [60]. Interestingly, a very recent study analyzed the effect of moderate weight loss on the rhythmic characteristics of the daily synthesis of leptin and ghrelin; the data demonstrated that although weight loss has been shown to reduce leptin serum levels, the rhythmic properties were similar in obese subjects underwent a hypocaloric dietary intervention for 12 weeks compared to normal-weight controls, suggesting that losing weight restores the daily rhythms of daily leptin synthesis [61].

Recently, the circadian rhythm of blood pressure has also been investigated in mice and humans [62]. Nocturnal dipping of blood pressure is part of the normal circadian pattern, and its absence (“non-dipping”) is more frequent in hypertensive patients [63]. Several lines of evidence suggest that shift work that causes disruption of the blood pressure circadian rhythm was significantly associated with metabolic syndrome [62]. As described in the literature, the ANS is increasingly recognized as an important pathway that mediates circadian variation in blood pressure [63]. Therefore, these data suggest that the normal circadian pattern of blood pressure may be altered by the dysregulation of ANS, likely due to a dysfunction of leptin circadian rhythm.

## 5. Conclusions

As reported in this review, the ANS plays an important role in the regulation of blood pressure but also in the regulation of body weight, satiety and energy homeostasis. In particular, we highlight the pivotal role of leptin in increasing energy expenditure, acting both on the cardiovascular system and BAT thermogenesis through the activation of SNS. 

In obese subjects, the sympatho-excitatory effects of leptin on the cardiovascular system are maintained while its metabolic effects result ineffective, suggesting that some form of obesity may be characterized by a “selective leptin resistance”. Therefore, in obesity, the failure in leptin action may lead to a reduction of energy expenditure and contributes to weight gain; at the same time, a compensatory hyperleptinemia may favor the development of hypertension and lead to overt cardiovascular disease.

Furthermore, the interplay between insulin and leptin plays a relevant role in glucose homeostasis and arterial blood pressure; therefore, a dysfunction of insulin and leptin crosstalk could be related to the alterations of the sympathetic activity and responsible for the subsequent development of hypertension and/or T2D.

Moreover, it has been highlighted that disruption of the circadian clock may alter leptin circadian rhythm and synthesis and may induce leptin resistance, causing the impaired regulation of metabolism and energy balance and promoting obesity and metabolic complications. These data place leptin as a major bridge linking circadian control and energy homeostasis.

However, the effects of leptin on several neural circuits, including the SNA, require further investigations in order to obtain a more complete picture of the mechanisms involved in the pathogenesis of obesity and its complications.

## Figures and Tables

**Figure 1 ijms-22-05187-f001:**
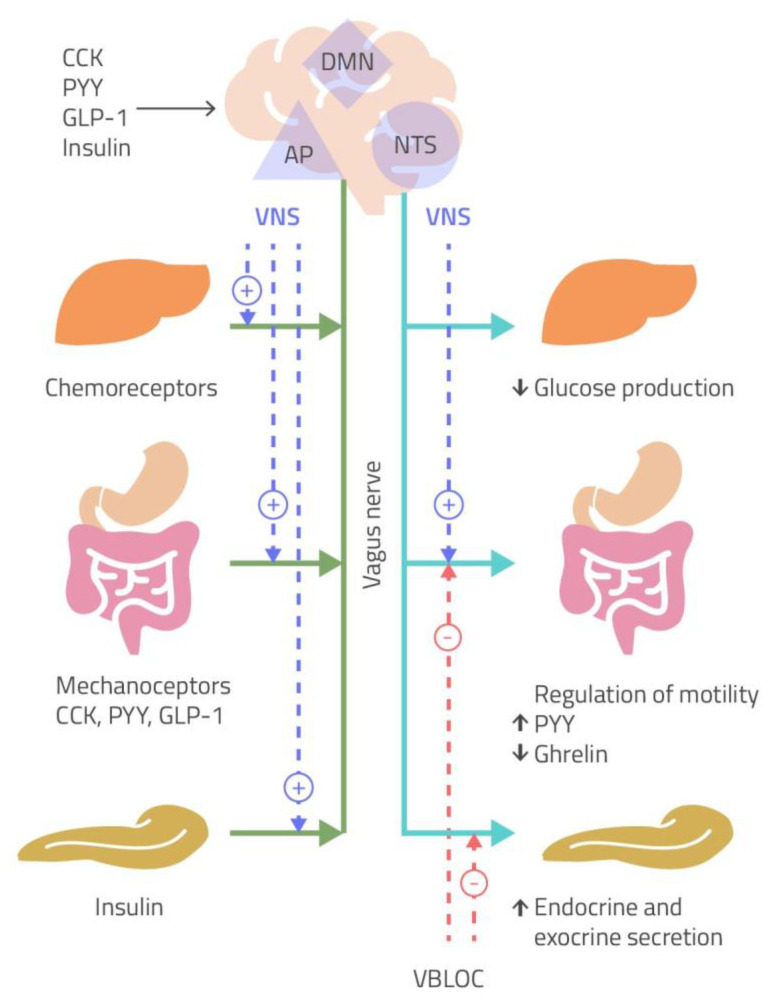
Peripheral signals of sense of satiety, after gastric distension, reach the nucleus of the solitary tract/area prostrema complex (NTS/AP) via afferent vagal nerves (green line). The NTS projects to the dorsal motor nucleus (DMN). This pathway modulates glucose production, gastrointestinal motility and hormone release (cholecystokinin, CCK; peptide YY, PYY; glucagon-like peptide-1, GLP-1), and pancreatic secretion via efferent vagal nerves (light blue line). The suggested site of action of vagal nerve stimulation (VNS) is indicated by the dotted purple lines, while the mechanism of weight loss hypothesized vagal nerve blockade (VBLOC) includes a decrease in gastric emptying and inhibition of pancreatic exocrine secretion (dotted red lines) [5].

**Figure 2 ijms-22-05187-f002:**
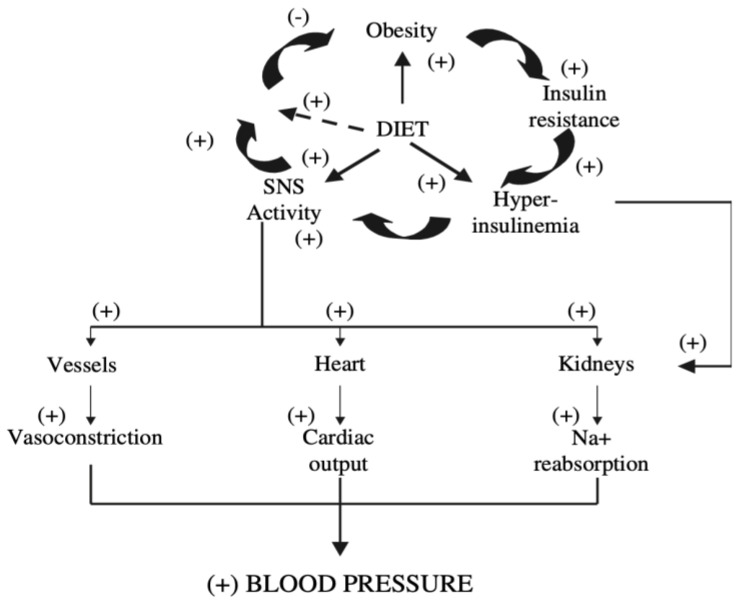
Relationship between obesity and blood pressure (BP). Insulin-mediated sympathetic stimulation is a mechanism recruited in the obese to restore energy balance by increasing metabolic rate. The increased BP is related to the increased levels of insulin and sympathetic nervous system (SNS) activity [3].

**Figure 3 ijms-22-05187-f003:**
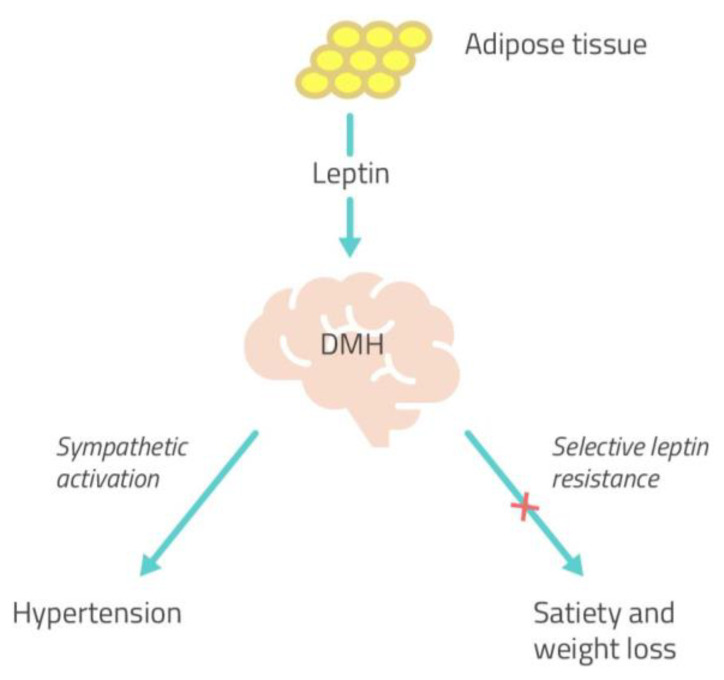
The concept of selective leptin resistance. The sympatho-excitatory effects of leptin on the cardiovascular system are maintained through its action in the brain region of the DMH, while its metabolic effects result ineffective, suggesting that some form of obesity may be characterized by a “selective leptin resistance” [36]. DMH, dorsomedial hypothalamus.

**Figure 4 ijms-22-05187-f004:**
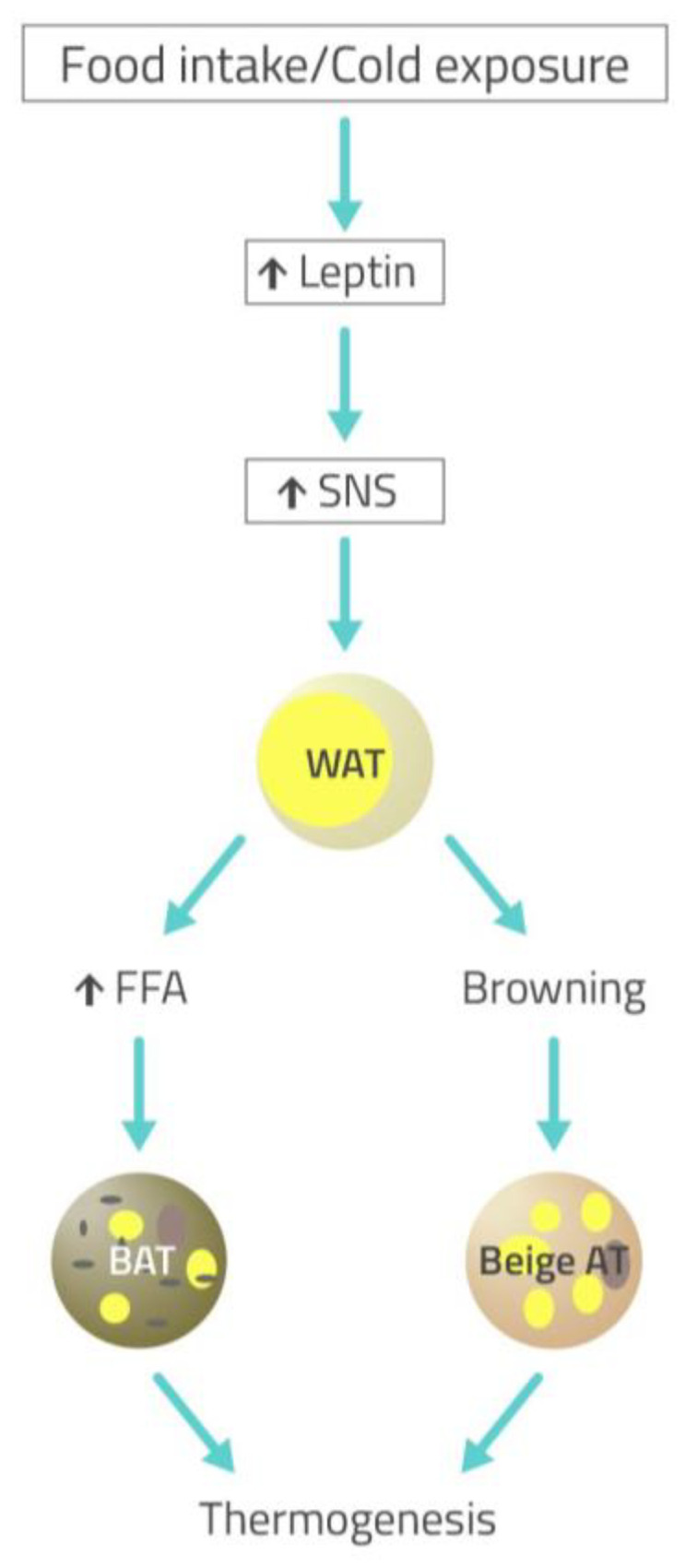
Cold exposure- or food intake-stimulated sympathetic activation through leptin action results in lipolysis in white adipose tissue (WAT) and thermogenesis in brown adipose tissue (BAT) and beige fat (Beige AT). Sympathetic nervous system (SNS) activated by leptin results in mobilization from WAT of free fatty acids (FFA), which are then used by BAT inducing heat production. Sympathetic activation also induces the formation of Beige AT through the browning of WAT, which contributes to adaptive thermogenesis [5].

**Figure 5 ijms-22-05187-f005:**
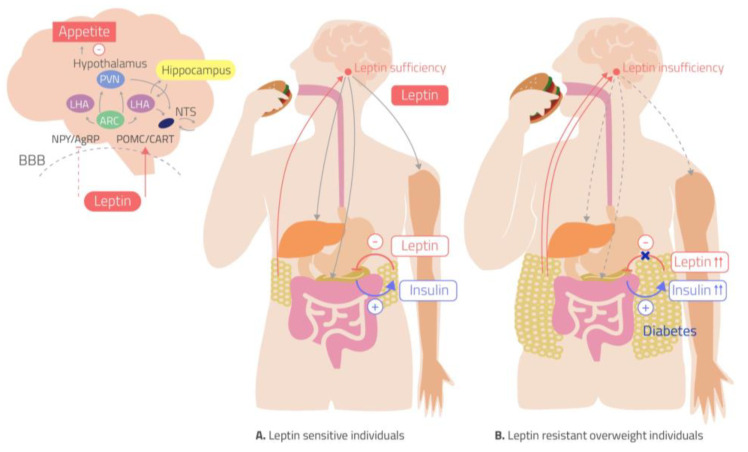
Leptin has an effect on appetite and insulin-glucose axis. In the hypothalamus, leptin activates pro-piomelacortin/cocaine-and-amphetamine responsive transcript (POMC/CART) neurons and inhibits neuropeptide Y/agouti-related peptide (NPY/AgRP) neurons, leading to anorexia. (**A**) In leptin-sensitive individuals, leptin inhibits insulin production and secretion from pancreatic beta cells while insulin stimulates leptin secretion from adipose tissue. Leptin increases glucose uptake in skeletal muscle tissue and stimulates liver insulin sensitivity via the sympathetic nervous system (SNS). (**B**)The leptin-resistant overweight individuals are resistant to the anorectic and weight-reducing effects of leptin, despite the increase in plasma leptin levels. Leptin resistance leads to hyperinsulinemia which, in turn, increases plasma leptin [45]. ARC, arcuate nucleus; LHA lateral hypothalamic area; NTS nucleus of the solitary tract; PVN, periventricular nucleus.

**Figure 6 ijms-22-05187-f006:**
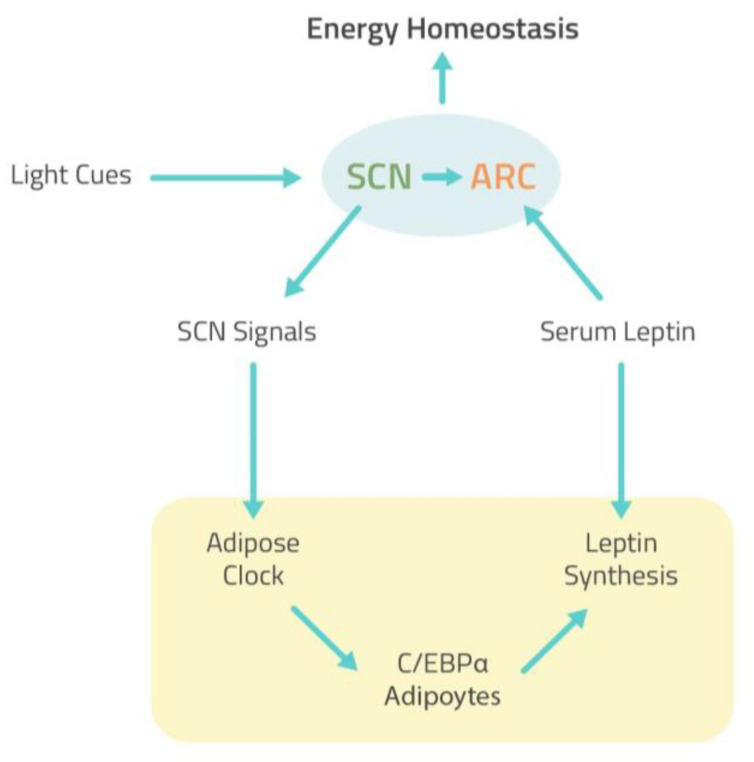
The leptin endocrine feedback loop to maintain the homeostasis of energy balance. In the CNS, the SCN clock rhythmically transmits the signal to the peripheral adipose tissue clock and potentiates the response of ARC neurons to circulating leptin. In white adipocytes, the BMAL1/CLOCK heterodimer that received signals from SCN directly controls leptin expression by regulating the activity of C/EBPα. Leptin stimulates SNS activity via the ARC of the hypothalamus to maintain energy homeostasis [54]. CNS, central nervous system; SCN, suprachiasmatic nucleus; ARC, arcuate nucleus; C/EBPα, CCAATenhancer-binding protein alpha; SNS, sympathetic nervous system.

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
