# Peer review of "Autonomic Nervous System in Obesity and Insulin-Resistance—The Complex Interplay between Leptin and Central Nervous System"

_ijms, 2021, doi:10.3390/ijms22105187_

Round 1
Reviewer 1 Report
The presented manuscript concerns a very important issue, which is the regulation of organisms energy homeostasis as well as dysfunctions such as obesity and insulin resistance and the mechanisms responsible for their etiology. This question is studied in many laboratories, but because of the complexity of the regulatory systems involved in this issue is still not fully resolved.
A serious problem concerning obesity and obesity-related diseases has recently emerged in civilized societies causing increased morbidity and mortality. Eating disorders are often related to disturbances in neuroendocrine control of metabolic processes and brain neurohormone activity and may generate similar neurobiological pathologies resulting from neuroadaptive changes at the central nervous system (CNS) level
The presented review summarizes and systematizes the vast amount of information about leptin action, which has been the subject of intensive research in recent years. Taking into account the results of research carried out in the last 15 years, especially after the discovery of leptin, adipose tissue is no longer considered only as a storehouse of the body's energy reserves but is assumed to play an important role in the regulation of metabolism and also has the ability to synthesize many peptides with numerous regulatory functions. The article describes the complex relationship between leptin and the central nervous system, provides information on the role of the autonomic nervous system in the function of adipose tissue and the etiology of insulin resistance.
I found this article very interesting. The authors provide an accurate and clear summary of the current knowledge about leptin and its participation in the mechanisms regulating the maintenance of the body's energy homeostasis. An additional value of the article is the numerous carefully prepared figures. I think that these drawings illustrate very well the text presented and facilitate understanding. The article is based on a large bibliography (63 works), of which 14 are articles from the last 5 years
Author Response
We are very grateful to the reviewer for his/her positive evaluation of our paper.
Reviewer 2 Report
This review is focusing on how autonomic nervous system contributes to obesity and metabolism. The subject is interesting and worth reviewing. Descriptions are appropriate. A concern is that the content looks like that of ref. 5 (Guarino D et al, The Role of the Autonomic Nervous System in the Pathophysiology of Obesity. Front Physiol 2017, 8, 665). Especially, two figures are almost the same. It would be better if figures have more originality.
Author Response
We wish to thank the reviewer for his/her constructive comments. We agree with the reviewer that the two figures are very similar to those of ref 5. Following the suggestions of the reviewer, we therefore modified figure 3 (new figure 4), highlighting the action of leptin on SNS and the effect on adaptive thermogenesis. However, we decided not to modify figure 1 since it describes in a complete and comprehensive manner the peripheral signals of sense of satiety, related to vagal activity discussed on this review.
Reviewer 3 Report
In this review, the authors described the role of the autonomic nervous system (ANS) in obesity and insulin-resistant conditions, focusing on leptin as a major player. The authors discussed the role of leptin on appetite, energy expenditure and autonomic effect. Further, the link between leptin and circadian clock was also discussed in terms of glucose metabolism and energy balance.
Overall, the manuscript was written well and easy to follow. However, the role of leptin on energy balance through ANS discussed in this article has been already well appreciated, and this reviewer feels that this review adds only little progress in this field. The authors are recommended to explain the novelty and merit of this review more clearly. Especially, “selective leptin resistance” should be discussed more intensively.
Author Response
We wish to thank the reviewer for his/her constructive comments. Following the reviewer’s suggestions, the concept of “selective leptin resistance” is now discussed more intensively (line 146-170, page 4-5) and we added a new figure (figure 3) that illustrates potential consequences on hypertension and metabolic effects.
Round 2
Reviewer 3 Report
The authors have responded to the comments properly.